# Factors Associated with the Quality and Transparency of National Guidelines: A Mixed-Methods Study

**DOI:** 10.3390/ijerph19159515

**Published:** 2022-08-03

**Authors:** Tanja Kovačević, Davorka Vrdoljak, Slavica Jurić Petričević, Ivan Buljan, Dario Sambunjak, Željko Krznarić, Ana Marušić, Ana Jerončić

**Affiliations:** 1Department of Pediatrics, University Hospital of Split, 21000 Split, Croatia; tkovacevic@kbsplit.hr; 2Department of Pediatrics, University of Split School of Medicine, 21000 Split, Croatia; 3Cochrane Croatia, University of Split School of Medicine, 21000 Split, Croatia; davorka.vrdoljak@mefst.hr (D.V.); ibuljan@mefst.hr (I.B.); ana.marusic@mefst.hr (A.M.); 4Department of Family Medicine, University of Split School of Medicine, 21000 Split, Croatia; 5Department of Pulmology, University Hospital of Split, 21000 Split, Croatia; slavica.juric01@gmail.com; 6Department of Research in Biomedicine and Health, University of Split School of Medicine, 21000 Split, Croatia; 7Center for Evidence-Based Medicine and Health Care, Catholic University of Croatia, 10000 Zagreb, Croatia; dsambunjak@gmail.com; 8Department of Internal Medicine, Division of Gastroenterology and Hepatology, University Hospital Centre Zagreb, 10000 Zagreb, Croatia; zeljko.krznaric1@zg.ht.hr; 9Croatian Medical Association, 10000 Zagreb, Croatia; 10University of Zagreb School of Medicine, 10000 Zagreb, Croatia

**Keywords:** knowledge, education, public health, national guidelines, guideline development, clinical practice guidelines, methodological quality, focus group

## Abstract

We assessed the methodological quality and transparency of all the national clinical practice guidelines that were published in Croatia up until 2017 and explored the factors associated with their quality rating. An in-depth quantitative and qualitative analysis was performed using rigorous methodology. We evaluated the guidelines using a validated AGREE II instrument with four raters; we used multiple linear regressions to identify the predictors of quality; and two focus groups, including guideline developers, to further explore the guideline development process. The majority of the guidelines (N = 74) were developed by medical societies. The guidelines’ quality was rated low: the median standardized AGREE II score was low, 36% (IQR 28–42), and so were the overall-assessments. The aspects of the guidelines that were rated best were the “clarity of presentation” and the “scope and purpose” (median ≥ 59%); however, the other four domains received very low scores (15–33%). Overall, the guideline quality did not improve over time. The guidelines that were developed by medical societies scored significantly worse than those developed by governmental, or unofficial working groups (12–43% per domain). In focus group discussions, inadequate methodology, a lack of implementation systems in place, a lack of awareness about editorial independence, and broader expertise/perspectives in working groups were identified as factors behind the low scores. The factors identified as affecting the quality of the national guidelines may help stakeholders who are developing interventions and education programs aimed at improving guideline quality worldwide.

## 1. Introduction

At a national level, evidence-based medicine (EBM) is introduced into clinical practice primarily through clinical practice guidelines (CPGs) [1]. CPGs include “recommendations to optimize patient care based on a systematic review of evidence and an assessment of the benefits and harms of alternative care options” [2]. They have been found to improve treatment outcomes, the cost-effectiveness of a healthcare system, and the maintenance of its quality standards [3,4,5,6]. However, to achieve the goals of improved resource utilization and health care delivery, CPGs must be high-quality in both content and form. Poor-quality CPGs have been shown to negate their potential benefits and can lead to suboptimal, ineffective, or harmful practices [7,8]. Still, because of the complexity and the demanding nature of guideline development (GD) and guideline implementation (GI) processes, the quality of the evidence-based information that is translated into clinical practice varies widely [9,10,11,12,13].

The development of high-quality CPGs, including de novo, adopted, adapted, or contextualized guidelines, is required for successful clinical practice guideline implementation and has been identified as critical in closing the gap between international recommendations and realistic best practice in low- and middle-income countries [14]. Studies that have examined the quality of CPGs for a specific clinical topic have found that it varies considerably between countries [9,11,15,16,17,18,19]. However, to gain an insight into the overall quality of national CPGs, a representative sample of national CPGs (all, or a random sample of the CPGs published within a certain time period) should be assessed. So far, few studies have examined an entire body of national guidelines, and they have found that countries, on average, produce either low- [20,21,22,23] or high-quality guidelines [24,25]. This means that in countries where guideline developers produce typically poor-quality guidelines, the chances of providing evidence-informed and cost-effective health care are significantly reduced. This is especially concerning given that it is often necessary to make recommendations at a national level due to the uniqueness of each country’s healthcare system—in epidemiological, social, organizational, economic, and political contexts—and due to different patients’ expectations across countries, making the use of international guidelines in such cases insufficient.

While numerous studies have investigated the barriers and facilitators of the guideline implementation process, the factors influencing the guideline development process are relatively underexplored [26], particularly at a country level.

The mapping of factors that are potentially affecting the global guideline development process, which were inferred from qualitative studies that discussed and evaluated GD steps, revealed that flaws in the process—such as a lower rigor of evidence synthesis; the omission of a transparent and usable summary of evidence, made available for discussion; flaws in considering the evidence, balancing it with panel members’ input, and opportunities to interpret the evidence; inappropriate panel member selection or conduct; and conflicts of interest, among other factors—may contribute to the lower quality of a CPG [27]. At a country level, only one study has been conducted, which was in India. This study focused on the GD of a specific set of CPGs and revealed several barriers of the process, in the form of attitudes towards the use of evidence, a lack of methodological capacity, an inadequate governance structure, and funding [28].

Few studies have looked at the correlations between the quality of global and national CPGs, covering a wide range of health-care topics—as measured by the Appraisal of Guidelines for Research and Evaluation II tool (AGREE II) and the guideline characteristics. Burgers et al. studied the CPGs published between 1992–1999 and discovered that the quality of the AGREE domain, “rigor of development”, was related to the level of care (primary/secondary or tertiary/all levels), the scope (prevention/diagnosis, treatment, combination), and the guideline program (part of the program/not part of the program) of the CPGs [29]. The authors also observed that the guidelines developed by government-funded agencies had the highest overall scores across all the domains, although statistical significance was reached only for the “editorial independence” domain. They additionally observed that time had no effect on the AGREE domains, except in the “clarity of presentation” domain. On the other hand, Alonso-Coello et al. examined the CPGs published between 1980–2007 and discovered that time had a significant effect on the quality of all the AGREE II domains, except for the “editorial independence” domain, and that geographic region had an effect on the quality of the “scope and purpose” and the “clarity of presentation” domains [12]. They also found that medical societies scored lower than governmental and international institutions in all domains, except “editorial independence”. Finally, Armstrong et al. assessed the CPGs published between 1992 and 2014 and found a significant improvement in the quality over time in all of the domains of the AGREE II tool [30].

With a few exceptions, studies that have assessed the quality of a representative sample of national CPGs did not examine the factors influencing their quality [20,21,22,24,31,32,33,34]. In Argentina, the quality of the national CPGs produced between 1994 and 2004 did not change over time and was not related to the type of guideline dissemination (published vs. not published); the level of healthcare provided by the organization involved in the GD (macro—Ministry of Health; meso—individual health care providers, organizations of providers, and health insurance institutions; or micro—individual health professionals); the type of guideline (prevention vs. treatment/diagnosis CPG); or the scope (national vs. regional/local guidelines) [23]. However, the authors did find that the guidelines produced by institutions belonging to more than one level of the healthcare system had higher scores than the guidelines produced by institutions belonging to only one level, and that prevention CPGs had higher scores than the treatment/diagnosis management guidelines. In China, the authors determined an increase in quality over time for the guidelines produced between 2014 and 2018, with subgroup analyses revealing that both the methodological and reporting quality were generally higher among the CPGs that used evidence grading systems or reported receiving funding [35]. Finally, in Japan, the quality of all the AGREE II domains improved over time for the guidelines produced between 2000 and 2014 [24], whereas for those published between 2018 and 2019, the effect of time was not tested, but the authors discovered that the number of guideline developers, the clinical question ratio, and the adopted guideline development methods (GRADE and Minds vs. others) were significantly related to the quality of the “rigor of development” domain [36].

Identifying the factors that systematically affect the overall quality of national guidelines and, consequently, the quality of health care in a country is of both national and international importance. Such knowledge would provide a solid foundation for identifying the issues in the process of developing CPGs that are shared among different countries, and would help to improve the quality of future guidelines [37]. To gain this knowledge, an in-depth, systematic investigation is needed, which not only assesses the quality of the CPGs in a country by using a representative sample, but also investigates the possible reasons for such quality by interviewing the country’s guideline developers and users, and examines which CPG characteristics are associated with the level of quality.

The main objective of our study was to assess the methodological quality and transparency of all the CPGs published in Croatia up until 2017 and to systematically investigate the factors affecting this quality using a quantitative regression analysis, with guideline characteristics as predictors, and a qualitative analysis with two focus groups, comprising guideline developers and users.

## 2. Materials and Methods

### 2.1. Quantitative Study

We aimed to identify all CPGs developed in Croatia up until 2017 and found 81 documents published as CPGs in the Croatian Medical Association’s (CMA) official journal, *Liječnički vjesnik*, which is a target journal, publishing all Croatian CPGs [38]. Seven documents were excluded because they were narrative reviews, clinical pathways, translations of foreign guidelines, or training manuals.

We used the AGREE II tool to assess the methodological quality and transparency of guidelines [39,40]. The tool was released in 2009 [39], and is now the most widely used and verified guideline evaluation tool. It is comprised of 23 items, grouped into the following six domains: “scope and purpose”, “stakeholder involvement”, “rigor of development”, “clarity of presentation”, “applicability”, and “editorial independence”. It is rated on a 7-point scale (from 1—“strongly disagree”, to 7—“strongly agree”). Based on this ranking, a standardized domain score, ranging from 0% to 100% (low- to high-quality), is computed for each of the six domains. Standardized overall score (0–100%) can also be reported [24]. AGREE II has two additional global ratings—overall assessments. The overall quality of the guidelines is scored on a 7-point scale (1—“lowest possible quality”, to 7—“highest possible quality”), while the second evaluation provides a recommendation on the guideline use (“yes”, “yes, with modifications”, “no”). In this study, each of the four raters independently rated all 74 CPGs. For more details see Methods—Quantitative study, Appendix A.

Categorical data distributions were presented with absolute numbers and percentages, and quantitative with median and interquartile range (IQR), due to asymmetrical distribution. Multiple linear regressions were used to determine predictors of CPG’s quality and transparency. The standardized AGREE II scores (overall, and domain scores) were dependent variables in these models, while independent variables were as follows: type of organization leading GD, number of entities/organizations involved in GD, and the year of publication. Conclusions were drawn at a significance level of 0.05, using two-sided tests. Confidence intervals (CIs) of 95% were also considered. SPSS 24.0 software (IBM Corp., Armonk, NY, USA) was used for statistical analysis.

The VOSviewer software was used for visualization of the co-citation network of Croatian guideline authors [41].

#### Comparison with Other Countries

We conducted a systematic search in PubMed to find studies that investigated the quality of national CPGs based on a representative sample, using the AGREE tool. Appendix A (Methods—Quantitative research) contains the search filter and details on inclusion and exclusion criteria. If the publication period of the analyzed CPGs included more recent years—2012 and later—median standardized domain scores for a country were extracted from each domain for that study and plotted on the radial graph.

### 2.2. Qualitative Study

To further investigate factors affecting the quality of Croatian guidelines, we conducted two focus groups, including the following: guideline developers (N = 6) and potential guideline users (N = 5). In total, 10 physicians and 1 dentist took part, working for primary, secondary, and tertiary health care organizations or Croatian Insurance Fund. Interviews were conducted by DS at the Psychology Research Laboratory of the Catholic University of Croatia, in Zagreb. The sessions, which lasted 1 h, were audio recorded. The conversations in both groups started with the question, “Why are clinical guidelines important in Croatian healthcare?”, after which we let the discussion run its course. We had prepared additional questions in case of a silence, but the discussions were lively.

The discussions were then transcribed into an electronic document and analyzed using grounded theory approach. We used an iterative process for reading the answers, open coding, and constant comparison to generate themes and patterns [42]. For more details see Methods—Qualitative study, Appendix A. The checklist for reporting guideline COREQ is provided in Appendix A.

## 3. Results

### 3.1. Quantitative Study

Seventy-four Croatian CPGs were published up until 2017. The number of published CPGs increased linearly with time, from just one (1%) CPG published in 2004, to 12 (16%) in 2017 (see Appendix A).

The development of the CPGs was led by CMA’s medical societies (81% of the 74 guidelines), the Ministry of Health working groups (WGs) (7%), and unofficial WGs, comprising physicians-volunteers, and, in one case, a patient association (12%). Five out of 163 CMA societies [43] (two different oncology societies and societies for gastroenterology, urology, and clinical nutrition) were involved in the development of 43% of the guidelines. The authorship landscape of the Croatian guidelines demonstrates considerable interconnection, with eight clusters apparently topic-oriented (three belonging to oncology topics and one to gastroenterology) and some strongly connected authors that also connect to other clusters (Figure 1).

The CPGs most often targeted neoplasms (38%), followed by endocrine (8%), and infectious (8%) diseases. None of the CPGs focused on children’s health, and no guideline was developed for patients.

Only seven guidelines (9%) envisaged updates, either in a three- or five-year period, or in the case of new scientific evidence. Still, only one guideline from 2011 was updated in 2017, and this update did not specify whether a new update is planned.

Regarding the methodological quality and transparency of Croatian CPGs, the median standardized overall AGREE II score was 36% (IQR 28–42%), with all the guidelines obtaining less than 50%, except for seven (9%) guidelines, which achieved between 63–76%. However, the domains differed considerably in the level of quality that was assessed (see 95% CIs in Appendix A), with “clarity of presentation” and “scope and purpose” scoring by far the highest (the median standardized domain scores were 74% and 59%, respectively) and “applicability” scoring the lowest (15%). “Rigor of development”, “editorial independence”, and “stakeholder involvement”, with medians of 23–33%, were comparable (Figure 2A; Appendix A). In terms of the variability in the quality assessment, the “applicability” and “rigor of development” ratings were lower and more homogeneous than those of the other domains.

The guidelines’ overall AGREE II assessments on the 7-point scale were mainly rated low to moderate (76% of CPGs received up to four points, see Appendix A). Overall, the raters did not recommend 31 (42%) of the guidelines, they recommended 38 (51%) with modifications, and only five (7%) were recommended for use in clinical practice.

#### 3.1.1. Comparison with Other Countries

Figure 2B shows a radial graph comparing the standardized AGREE II domain scores in Croatia to the scores in other countries, where studies analyzed the whole body of national guidelines published during a specified period, assessed the quality using the AGREE II instrument, and reported the domain scores. The findings in Croatia were comparable to the findings in other countries. Except for Japan—which has high-quality guidelines and scored above the cutoff of 60% on the “rigor of development” domain as well as the “clarity of presentation” and “scope and purpose” domains—all other countries developed, on average, low-quality guidelines that followed the same pattern as Croatia. The highest scoring domains were “clarity of presentation” and “scope and purpose”, which were only actually rated as low-quality in those countries with, on average, extremely low overall AGREE scores (<30%). The domain with the lowest score was “applicability”, which was followed by “rigor of development” and “editorial independence”.

#### 3.1.2. Predictors of Methodological Quality and Transparency

The predictor of the methodological quality and transparency of the guidelines, measured by the AGREE II standardized score—overall and per domain—was the type of entity/organization that led GD, not the total number of participating organizations (Table 1). Compared with the guidelines developed by medical societies, the guidelines developed by a governmental entity performed 22–43% better in all the domains of AGREE II. The guidelines developed by unofficial WGs/non-governmental organizations performed 12% better in the “rigor of development” domain. These latter guidelines also appeared to perform better than those produced by medical societies in the areas of “clarity of presentation” and “applicability” by 7–8%, as assessed at the 0.1 significance level. The year of publication was a weak predictor for the domains “stakeholder involvement” and “editorial independence”, increasing 2–3% per year. For all other domains and for the overall score, the quality did not change over the time period (Table 1, Figure 3).

### 3.2. Qualitative Study

In Table 2, we list the themes identified in the focus group discussions, with representative quotes for each topic. In the paragraph below, we describe the qualitative findings in detail.

#### 3.2.1. Guideline Development—Framework

When discussing which entity should lead the development of Croatian guidelines, both developers and users preferred a bottom-up approach, with medical societies as CPGs’ main developers. One developer stated that even if the topic of the guideline and the call from it came from the Ministry of Health (top-down approach), it was still medical societies that should answer that call and develop them. Reference centers were not seen as feasible in the Croatian system by either developers or users, due to the inability to have a sufficient number of centers and experts to cover all the necessary areas of care in a small country such as Croatia. One top-down approach that was believed possible was the establishment of a dedicated national agency, responsible for developing guidelines.

The topic of prioritizing GD emerged among developers but was not discussed extensively. One developer believed that the criteria for decisions about which CPGs to develop should be based on disease prevalence and another believed that the criteria should be based on affordability and the complexity of the guidelines.

#### 3.2.2. Guideline Development—Composition of Working Groups

Concerning the composition of a WG for GD, users preferred a combination of “medical societies and medical experts”. They also identified the need for a broader range of experience and perspectives, listing as prospective WG members, various medical specialists and insurance company representatives, or patients. One guideline developer stressed the need to have “an expert”, with experience in the methodology of GD.

#### 3.2.3. Guideline Development—Methodology

Participants from both groups identified inadequate methodology as a barrier to the development of high-quality CPGs. When developers described the steps that WG members normally follow, they revealed that, commonly, no systematic, evidence-based methodology was used, and that CPGs were based on expert opinion.

A lack of knowledge was seen as the major reason for weak methodology. While describing the GD process, two developers associated evidence only with the existence of meta-analysis and considered only the strength of evidence and pharmacoeconomic considerations as important factors for making recommendations. Furthermore—in opposition to the order recommended by the Grading of Recommendations Assessment, Development and Evaluation (GRADE) handbook [44], and those cited in the AGREE II tool [39]—they stated that pharmacoeconomic considerations should be considered before assessing the strength of evidence, or they confused the GD with the GI steps—they described that when developing a guideline, developers first determine which “process does not ‘breathe’ well” and then identify “critical points in it”, pointing out that the methodology of guidelines is to eventually establish which “element (clinical knowledge + resources + preferences/values) is dominant”. However, it is a predefined clinical question, not a process, for which one should assess the quality of evidence and decide on the strength of the recommendation by considering all the factors: health benefits, side effects, and risks.

One developer of expert-opinion guidelines believed that their colleagues did a very good job methodologically. A user further stated that formal education or specialized training is not needed for GD, because anyone can self-educate using free (online) resources (Table 2, Knowledge on GD Methodology).

### 3.3. How to Improve the GD Process in Croatia

Developers suggested several approaches to improve GD in Croatia, including the following: increasing the level of knowledge of GD methodology among members of WGs, increasing the level of motivation of the clinicians involved, determining who should lead GD, mapping all CPGs currently in use before prioritizing the GD of new guidelines, and allocating enough resources in terms of time and money.

### 3.4. Definition and Purpose of CPGs

Participants had different definitions regarding what a CPG is. While some identified them as evidence-based documents, others defined them as being sets of rigid or formal rules, standard operating procedures (SOPs), and even clinical pathways. Users also perceived them to “limit the freedom” of practitioners, calling them “*cookbook*”, “*traffic control*”, or “*just burdens that could not make a difference in their work environment*”.

Participants in both groups identified establishing a standard of care as the primary purpose of CPGs, but they also perceived CPGs to mitigate physicians’ conflict of interest in treatment selections, transfer decision-making responsibilities from physicians to the guidelines, and to educate physicians on optimal patient management. Some also expressed that CPGs are intended to provide some form of quality control, to improve the control and optimization of the healthcare business processes, or to lead to a better control of costs or payment billing.

## 4. Discussion

Our analyses have shown that the first Croatian guideline was developed in 2004, 13 years after Croatia’s independence, and, since then, the number of guidelines has steadily increased, reaching 74 in 2017. As there was not, and still is not, a central body overseeing GD in Croatia, national guidelines were most often developed by medical societies, grouped under the CMA.

The overall methodological quality and transparency of the national guidelines was poor. The standardized median overall score of AGREE II was 36% and almost all guidelines scored less than 50%, leaving much room for improvement. Tudor et al. studied seven Croatian neurological guidelines, which were published up until 2012, using the AGREE II instrument [45] and also reported that the guidelines largely lacked a structured development process and had a poor quality in the following domains: “applicability”, “editorial independence” and “stakeholder involvement”. Tokalic et al. found inadequate reporting of 24 Croatian and relevant transnational CPGs published between 2014 and 2016, using the RIGHT reporting checklist [46].

In line with other countries, the median standardized domain scores for “clarity of presentation” and “scope and purpose” were the highest, and their quality was deemed acceptable. In fact, these domains only received a low-quality rating from countries that were producing CPGs with an extremely low overall AGREE II score (<30%): China, Mexico, and Peru [20,33,34,35]. Other domain scores for Croatian CPGs were very low, with “applicability” ranking the lowest, followed by “rigor of development” and “editorial independence”. Similar results were observed in other countries that generally produced low-quality guidelines [20,21,22,33,34,35], regardless of the guideline framework used, indicating a common underlying problem. The largest difference between the countries producing high- and low-quality CPGs was observed in the “rigor of development” domain. Japan, the country producing high-quality guidelines, also had the “rigor of development” domain rated as high-quality (median standardized domain scores of 62%) [24], which was surpassed only by Australia, whose average rating for 15 guidelines following the GRADE methodology was 71% [25].

### 4.1. Change in Quality over Time

Except for “stakeholder involvement” and “editorial independence”, which have grown by between 2–3% every year, the ratings for the AGREE II domains had not changed over the past 13 years. In Argentina, another country that produced poor-quality CPGs between 1994 and 2004, the methodological quality has not improved in any AGREE II domain, despite an increase in the number of guidelines over time [23]. In contrast, in China, the authors reported an increase in quality over time for the guidelines produced between 2014 and 2018, from 5% (“clarity of presentation”) to 13% (“scope and purpose”) [35]. A systematic review of the studies that aimed to evaluate the CPGs published globally between 1980 and 2007 found that their overall quality improved with time. However, only the ratings for the following three AGREE II domains have increased since 2003, when the instrument was first published: “scope and purpose”, “stakeholder involvement”, and “clarity of presentation”. However, the scores for “rigor of development”, “applicability”, and “editorial independence” remained unchanged. The authors were primarily concerned by the fact that the mean score for “rigor of development” was just 43% (95% CI 41–45), as they considered this category to be the most important indicator of the guidelines’ methodological quality. A fundamental issue, according to the authors, was a scarcity of methodological professionals and methodological expertise so they recommended that de novo guidelines should only be developed when there are minimal requirements for them [12]. A considerable improvement in the quality of the “rigor of development” domain in Japan has been linked to advances in social guideline infrastructure, rather than simply to the release of a GD handbook [24]. While the lack of a time trend in the quality improvement may be due to the short period in which published CPGs were observed, the absence of such a trend over longer periods in a country producing low-quality guidelines implies that activities aimed at improving the quality of national CPGs are either nonexistent or deficient.

### 4.2. Factors Affecting the Quality of Croatian CPGs

#### Guideline Framework

Our study found that even though medical societies led the bulk of GD, their guidelines were, on average, of a lower quality than those led by the Ministry of Health (worse on all domains) or informal WGs (worse only on the “rigor of development” domain). This is an interesting finding, as most of our focus group participants said they believe that medical societies should be leading national GD. Therefore, more education and guidance, for example, the inclusion of methodological experts, as one focus member suggested, but also of other experts, such as information specialists and economists, might be an approach to improve their quality. Another approach may be to simply have external experts evaluate the guidelines and suggest improvements before they are published.

While the findings from other countries that are generally producing low-quality guidelines show that simply choosing a national agency to oversee the GD process [20,22] over the medical/scientific societies [21,23] does not guarantee the production of high-quality CPGs, there is no doubt that the current GD framework should be improved further. Other nations’ experiences, such as those of Japan [24] and Australia [25], suggest that the simultaneous, coordinated actions of multiple organizations may increase the quality of national CPGs. In addition, as emerged during discussions, the issues that should be addressed at a framework level include the following: the prioritization of GD at a national level, which is currently suboptimal, preceded by the mapping of all the guidelines currently in use (guidelines published in the official CMA journal, and international guidelines that may or may not have been translated into Croatian, and, according to one participant, “guidelines that everyone tacitly follows but have never been formalized”); a resource distribution plan (time and finances); and greater clarity of responsibilities (“who is in charge”, “medical societies should be supported and motivated to lead… but GD should not be an obligation for them”).

### 4.3. Inadequate Methodology of GD

Consistent with the low AGREE II scores and the above-mentioned framework findings, developers and users repeatedly identified inadequate methodology as a significant barrier to the development of high-quality guidelines.

That the methodology was, indeed, inadequate was also revealed by the descriptions of a typical GD process given by developers throughout the discussion. These descriptions highlighted that WGs do not commonly follow a systematic, evidence-based methodology and that recommendations are mostly dependent on expert opinion. While experts may have a thorough comprehension of the available research base, they may also express opinions that are unsupported by independent evidence evaluations. Expert opinion alone risks offering selective, out-of-date, or biased perspectives of existing evidence [47,48]. Even in cases when there are evidence gaps, expert opinion should be gathered via rigorous qualitative research, such as Delphi surveys, to support the available evidence claims [49].

The lack of methodological knowledge among the WG members is supported by the quality of the Croatian guidelines, which were developed by utilizing approved AGREE or GRADE [44] methodologies (AGREE: four developed by Ministry of Health WGs and one by a medical society; GRADE: two developed by medical societies and two by unofficial groups). We showed that the utilization of these methodologies is not a guarantee of high-quality. All the Ministry of Health guidelines were rated high, including the one that did not use methodologies, while all the CPGs of the other organizations that used them, ranked low. That the handbook alone is not enough was also identified by Seto et al. [24].

As evidenced by statements about “Developers not being well-versed in the methodology…” and the misconceptions and overconfidence in methodological knowledge that we observed, a lack of methodological expertise among the developers surfaced as a likely explanation for the insufficient methodology. When the level of understanding of the topic is not adequate a fertile ground for knowledge overconfidence forms, which is commonly explained by the Dunning–Kruger phenomenon [50,51,52]. The phenomenon implies that less experienced persons are not entirely aware of the actual complexity of the topic being addressed and, as a result, they are unaware of how much knowledge is required to grasp all the aspects of the topic and, thus, consider themselves fully educated and trained before they are. Such cognitive bias may be a barrier to the future methodological training of developers, since, to engage in voluntary self-improvement it is important to acknowledge the gaps in one’s own knowledge and abilities, as well as to comprehend how one is seen by others [52].

Despite the participants acknowledging the lack of methodological knowledge, one of the users believed that EBM education during medical school or residency is not needed to acquire this knowledge as it could be acquired in other ways, while another believed the best way was to learn during the development and to have a methods expert in the working group. The question remains as to how this expert should be trained—perhaps a dedicated master’s program or online training, rather than training for all medical students.

The developers also cited a lack of motivation among the clinicians involved in GD, which they attributed mostly to a lack of financial incentives and because they viewed GD as a resource-intensive process, and a lack of resources in the form of time and money as reasons for the inadequate methodology.

### 4.4. Involvement of Other Experts and Stakeholders in Working Groups

In addition to the Croatian guidelines being rated poorly in the “stakeholders involvement” domain, we found that Croatian WGs were comprised nearly entirely of physicians and seldom of any other healthcare professionals. A lack of association between the quality of the guidelines and the number of participating organizations (reflecting the number of experts) shows that the amount of medical expertise is not sufficient for the development of high-quality guidelines.

While none of the participants fully acknowledged the importance of multidisciplinarity and diverse perspectives in working groups, the medical specialist-centric viewpoint appears to be slowly changing, as evidenced by the yearly increase in the AGREE II scores for the Stakeholder-Involvement domain and the comments about inclusion of patients (the reason for including patients was to “share responsibility,” not to broaden perspectives), or insurance company representatives in working groups. Still, the existing composition of WGs may have had an impact on the most critical step of GD—assembling an effective WG—by excluding stakeholders or experts, such as other healthcare professionals, other professions, patients and/or carers, and policy makers [53,54].

### 4.5. Editorial Independence

“Editorial independence” was also rated low by using AGREE II but has improved very slowly over time. Still, the topic did not emerge during the focus groups discussions. Other studies have found that national guidelines generally fail to reveal funding sources and conflicts of interest [12,24,55,56]. In countries with low-quality guidelines, the median standardized score for the domain ranged from 0–39%. This score was 39% in Japan, while in Switzerland, conflicts of interest statements were submitted in 44% and declarations of financial support in 29% of the Swiss CPGs published between 2008 and 2019 [12,20,23,24,55,57]. It should be examined further why the ethics of guideline development develops slower than the ethics of research publication [58].

### 4.6. Applicability

In Croatia there is no system for CPG implementation in health care. After publishing CPGs in the CMA’s journal and eventually providing some dissemination and education activities, there was no further implementation plan. We do not know how many of the published guidelines that Croatia attempted to implement, but, based on the focus group discussions, implementation attempts were rare and unsuccessful. However, this is not an isolated issue; a similar problem has been observed in Norway [59].

### 4.7. Methodological Knowledge, Misconceptions, and Overconfidence among Guideline Developers and Users

We also discovered several misconceptions regarding the definition and purpose of guidelines, which may have influenced both developers’ and users’ expectations, attitudes, and even educational interventions.

According to one user, “all EU rules are evidence-based”, as opposed to the Croatian ones, which are thought to be based on expert opinion. It is crucial to note, however, that evidence-based does not always imply high quality, as this is dependent on the available data [60].

Developers and users also confused CPGs with SOPs, or even clinical pathways. SOPs are tailored to specific work processes, equipment, or conditions and explain how to safely perform a repeatable work process or procedure, as directed. They are written in a step-by-step format for the workers performing the process, similar to a cookbook. This misconception may be reflected in participants’ descriptions of guidelines as “cookbooks” or “traffic regulations” that “place the decision in the hands of a physician”, and may have also led to the misconception that guidelines are a set of rigid and formal rules that must be followed. Unlike SOPs, which are limited to a specific process or procedure, CPGs are broad recommendations, which set out general principles that are open to interpretation and are applicable to a wide range of situations. Other studies have also found that clinicians believe that recommendations are categorical, prescriptive, and that they restrict professional practice, and have interpreted their findings to reflect a lack of subject knowledge and awareness of how guidelines are formed [61].

When it comes to the purpose of guidelines, developers identified the primary objective as establishing a standard of care to eliminate unwarranted variability in treatment, whereas users also recognized other purposes. Surprisingly, no one suggested improving patient care as a goal.

Some participants, including developers, have mistakenly assumed that the purpose of CPGs is control—control of the quality of care, and the control and optimization of healthcare business processes, costs, or even patient billing. While CPGs have the potential to improve a healthcare system’s cost-effectiveness and the maintenance of its quality standards [3,4,5,6], this is achieved indirectly, through successful GI. Such misconceptions may, however, cause high expectations regarding GI, which ultimately leads to disappointment and a loss of confidence in the concept of CPGs [62].

### 4.8. Limitations

The study’s limitation is associated with using the AGREE II tool. The tool assesses how specific items within various domains were reported, which may be biased if an item was included/performed during GD but not reported. Furthermore, we applied the commonly used standardized cut-off score of 60% [63,64,65], although no standard cut-off scores have been established to distinguish between high- and low-quality standards. Nonetheless, the tool has been validated in many clinical domains and by various specialists [66], and in all the guideline cases evaluated, information meant to be of high-quality was scored higher than the content designed to be of low-quality [66]. Furthermore, in our analysis, the median standardized scores for low-scoring domains clustered around low values, with a difference of approximately 20% between low- and high-scoring domains clearly distinguishing the groups.

We examined the guidelines up to 2017. Given that the guideline framework has not changed in the meantime, and that the COVID-19 pandemic temporarily impacted GD activity, we believe that the data reported accurately represent the state of the Croatian guidelines in a pandemic-free setting. Considering that a 14-year time span had little effect on the quality of CPGs in Croatia, and that the guideline framework has not changed, we do not expect the CPGs published in pre-pandemic 2018 (N = 6) and 2019 (N = 8) to have a significant impact on the results.

Another possible constraint of the study was the participants’ discontent with the day-to-day challenges of organization and resource management in the Croatian healthcare system, which required the moderator to constantly divert or refocus the focus groups. In this setting, more structured questions might aid with the identification of additional barriers. Nonetheless, the discussions accurately mirrored reality and supported many of our quantitative findings.

## 5. Conclusions

Croatia, like other countries that produce, on average, low-quality guidelines, demonstrated the comparable pattern of high- and low-scoring AGREE-II domains and almost no quality improvement over time, with “rigor of development” and “applicability” as the two lowest scoring domains. The poor ratings in these countries occur regardless of who leads their national GD—medical or scientific societies, a GD agency, or a department under the Ministry of Health—indicating a more fundamental problem with GD quality than the framework itself.

The factor that was associated with the low quality of the Croatian guidelines was the development of the guidelines by medical societies, as opposed to the Ministry of Health or unofficial WGs. In the focus group discussions, additional factors behind the low scores were as follows: inadequate methodology; a lack of implementation systems in place; and a lack of awareness of the need for editorial independence, or border expertise and perspectives in working groups. We also identified a lack of methodological knowledge, misconceptions, and overconfidence among developers, which may interfere with the outcomes of their future training.

Other nations’ experiences, such as Japan and Australia, suggest that efficient ways to improve the methodological quality of de novo, and, consequently, adapted and contextualized guidelines, may be achieved through the simultaneous collaborative actions of several organizations to build social infrastructure for guidelines, which would include the development of protocols for GD, methodological training, and an increase in guideline-related skills. Furthermore, the ongoing exposure of medical students to EBM and guideline topics during their studies may both raise the level of knowledge of the future users’ and developers’ and improve the evidence-based climate in the healthcare system.

## Figures and Tables

**Figure 1 ijerph-19-09515-f001:**
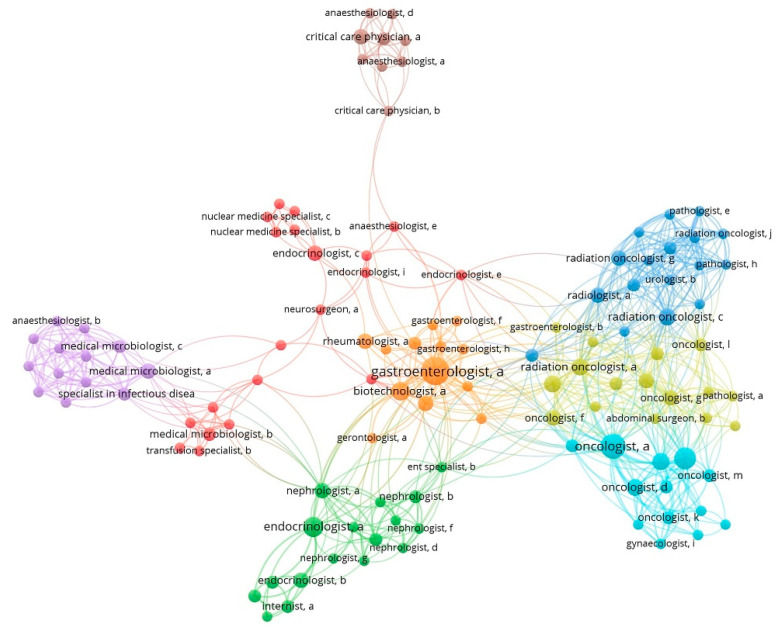
The authorship landscape of 74 Croatian clinical practice guidelines (649 unique authors, median—11 authors per guideline). Authors are identified by specialty, followed by an index letter after the comma that uniquely identifies a person. The size of the circle reflects the number of CPGs (≥2) published by this person.

**Figure 2 ijerph-19-09515-f002:**
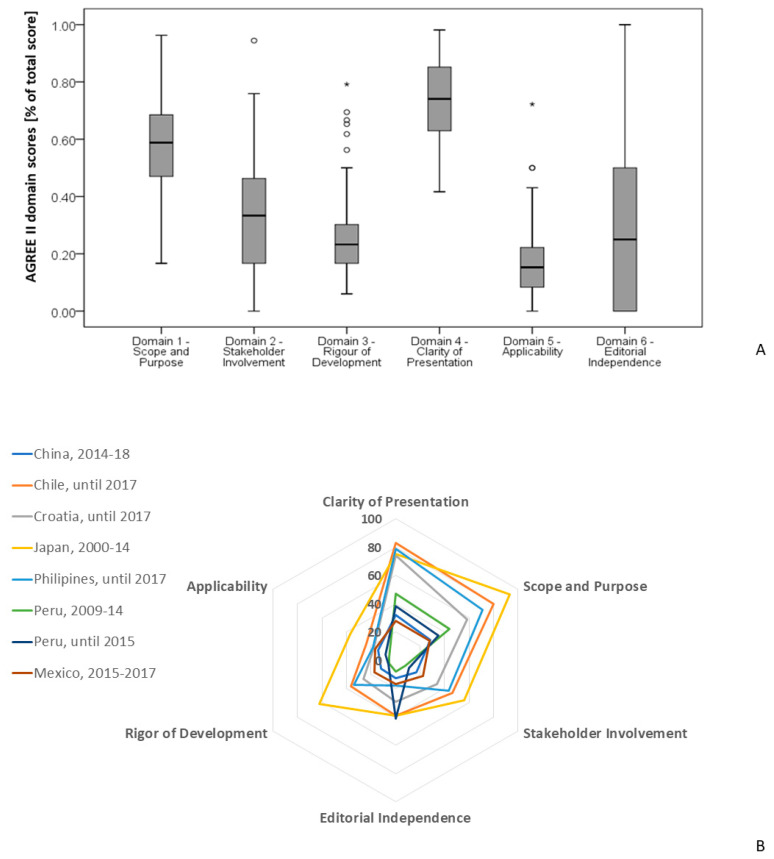
Standardized domain score distribution across six AGREE II domains: (**A**) among the Croatian clinical practice guidelines, N = 74. Outliers and extreme outliers are represented by circles and asterixis, respectively; (**B**) among countries that used AGREE II to evaluate national guidelines. The period studied and the country of origin are indicated in the legend of the figure. Japan was the only country reported to have high-quality guidelines [24], whereas all others were classified as those with low-quality guidelines [20,21,22,33,34,35].

**Figure 3 ijerph-19-09515-f003:**
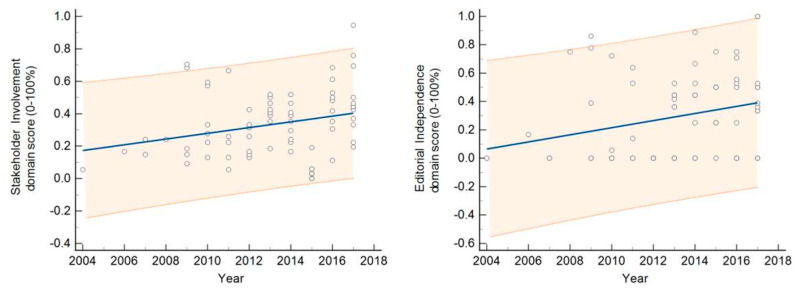
Croatian guidelines’ AGREE II scores in the domains “stakeholder involvement” and “editorial independence” increased over time. Shown here are the individual scores, as well as the simple regression line and its accompanying 95% prediction interval.

**Table 1 ijerph-19-09515-t001:** Effects of different factors on quality of Croatian guidelines, shown by AGREE II domains.

R^2^ = 20%	AGREE II: All Domains		AGREE II Domains											
Factors	Overall		Scope and Purpose		Stakeholder Involvement		Rigor of Development		Clarity of Presentation		Applicability		Editorial Independence	
	B *	*p*-Value	B	*p*-Value	B	*p*-Value	B	*p*-Value	B	*p*-Value	B	*p*-Value	B	*p*-Value
Year	-	0.292	-	0.522	0.02	<0.001	-	0.726	-	0.131	-	0.421	0.03	0.005
Total No. of entities	-	0.594	-	0.427	-	0.79	-	0.156	-	0.58	-	0.833	-	0.996
Entity primarily responsible for CPG’s development														
Led by unofficial WGs vs. led by medical society(ies)	0.08	0.017	-	0.699	-	0.138	0.12	0.002	0.08	0.065 †	0.07	0.050 †	-	0.466
Led by governmental health WGs vs. led by medical society(ies)	0.37	<0.001	0.37	<0.001	0.42	<0.001	0.43	<0.001	0.22	<0.001	0.35	<0.001	0.39	0.003

B—non-standardized regression coefficient, obtained by generalized linear models using year, total number of entities, and type of entities responsible for guideline development as independent variables; WGs—working groups. * Values for B are shown only for significant association; † association significant at 0.1 level.

**Table 2 ijerph-19-09515-t002:** Findings of the focus group discussions, organized by topic, with representative quotes by participants.

Topic	Examples of Quotes
**Guideline development framework—Who should lead GD?**	
Bottom-up approach	Medical societies, “It should be the medical societies.” Applications to a tender, “The Ministry of Health should issue a tender for the development of guidelines … Trendsetters and medical societies will apply…” Reference centers (judged as unfeasible in Croatian system), “There are too few reference centers for us to cover all areas of care… keep in mind that a reference center is made up of three, five, eight experts, and we have to think of all the other experts who work in other institutions.”
Top-down approach	“[national] agency dedicated to guideline development.”
**Guideline development framework—criteria for prioritization of GD**	
Disease prevalence	“If we start from patients who stand to benefit the most, we may end up developing guideline for only five patients.”
Expected costs and complexity of development	“Those are small guidelines… So, guidelines we can afford.”
**Guideline development framework—composition of a working group**	
Professional medical societies and medical experts	“Working group should consist of medical societies and experts”, while noting that “experts for certain diseases are easy to find.”
Diverse teams of various medical specialists and a representative of the insurance company	“Teams are important and a team developing clinical guidelines should consist of, for example, a diagnostician, a radiologist, an oncologist and possibly an insurance company representative. In my opinion, it would be good if they came from different centres.”
Experts and patients	“Maybe for this reason [in relation to the discussion about patients’ rights in Croatia, author’s note], patients should also be involved in the development of guidelines, so that they also understand the situation and take some of the responsibility.”
Other health professionals as part of a working group	[talking about making recommendations], “If it has elements to include some resources that are needed—then other professions participate in drawing conclusions too.”
**Guideline development methodology—methodology**	
Methodology is inadequate	“There are only a small number of guidelines that have been published so far in our country that have followed the correct methodological steps that should be followed in developing guidelines. … I would like to reiterate that most of our guidelines are methodologically inadequate.”
Description of a typical GD process in Croatia reveals that CPGs are mainly expert opinion-based	“When guidelines are developed, a group of people are brought together who are involved in a particular medical field or who work in some way with medical professionals in that field. For example, when guidelines were developed for the treatment of intraepithelial lesions of the cervix, gynecologists, pathologists, and cytologists were represented in this group, not only from Zagreb, but also from other centers. And then we met and discussed, each from the side of his profession, what to do in a particular situation. Of course, we did not make recommendations out of thin air, one part was based on our experience, another part was based on foreign experience that we had gathered at some meetings, congresses and so on. And that actually took quite a while before we as a working group published these guidelines.”
**Guideline development methodology—knowledge on GD methodology ***	
Developers lack methodological expertise	[developer], “Developers are not well-versed in the methodology of guideline development.”
Definition of GD process *	EBM and alignment with resources [user], “Guidelines should first be based on evidence-based medicine and then we should see how they can be aligned with our resources.” Evidence synthesis, pharmacoeconomic evaluation, strength of evidence, and consensus * [developer], “To develop a guideline, you need a meta-analysis, a systematic review for each diagnostic or therapeutic procedure, then you should include a pharmacoeconomic evaluation and then you should focus on strength of evidence and consensus. This process should be repeated several times until we agree on the final guidelines.” Clinical knowledge, resources, and preferences/values * [developer], “This implies that when we develop a guideline, we first identify which process doesn’t ‘breathe’ well for us, we identify the critical points in it, and then we determine if we have a problem at a critical point with: what is clinical knowledge, what some resources are, and what values or preferences we have. And then we get the take-home message that the methodology by which the guideline is created is actually to determine one of these elements that is dominant.”
Overconfidence on methodological knowledge **	[developer] Judgement about the methodological quality, made based on the recognition of a few topic-related elements by a developer of low-quality, expert opinion-based CPGs **, “I couldn’t physically attend all of these meetings [referring to working group meetings], but people who were more involved than me, in the sense that they knew what they were doing methodologically—I got the impression that they did a very good job and that they knew S1 and S2 and S3, so about the types of guidelines, when certain guidelines are issued, how much weight they have, and so on. So, while I didn’t have the task to do something methodologically, I got the impression that these people had done their homework thoroughly and knew what they were talking about.” [user] Underestimated complexity of the GD methodology **, “If someone asks you to develop a guideline, you will find a way because everything is available today.”
**How to improve GD process in Croatia**	
Increase the level of knowledge of GD methodology for members of WGs	“The people involved in the development of guidelines are not sufficiently familiar with the methodology of this process. I think that they need the help of an expert to explain this methodology to them to apply it correctly.”
Increase motivation of clinicians involved, possibly through financial incentives	“The GD process itself is very demanding. When we talk about the clinicians involved in it, they are not… how shall I say…? I am not going to say they should be paid for it; that may sound harsh. However, they are not motivated enough for such an action.”
Determine who should lead GD	“What is very important to say is who is in charge of writing these guidelines…” “Medical societies should be supported and motivated to lead GD in Croatia… GD should not be an obligation for professional societies—it would not work.”
Map all guidelines currently in use	“Before prioritizing the development of new guidelines, all guidelines currently in use in Croatia should be identified. This would require screening all hospitals and interviewing the experts in those hospitals to find out which international guidelines they already use.”
Allocate resources	“Resources—time and money should be provided.”
**Definition of clinical guidelines**	
CPGs are evidence-based	“First, our guidelines must be based on evidence-based medicine …”
CPGs are a set of rigid and formal rules that must be followed *	“…we believe that a guideline is set in stone.” *
CPGs are standard operative procedure documents *	“In our work, we use many clinical guidelines; only we do not call them that, we call them standard operative procedures.” *
CPGs are clinical pathways *	“You need to know the reason why you are moving a patient from a smaller hospital to a larger one.” *
**Purpose of clinical guidelines**	
To establish a standard of care	“So that we can ensure that every doctor in family medicine works at the same level, that pathologists read the findings the same way, and that rheumatologists treat the patient the same way.”
To mitigate potential risks arising from a physician’s conflict of interest in treatment selection	“Guidelines enable us to unify a system comprised of thousands of physicians with diverse personal interests.”
To move some of the responsibilities for patient care decision making away from physicians and toward guidelines	“Guidelines shift some of the responsibility for patient care decision-making from physicians to guidelines.”
To educate clinicians about the optimal management of patients	“If there were guidelines on how to treat a patient in certain circumstances, then people would be educated on how to solve the problem in that hospital…”
To achieve some level of control in the healthcare system *	Quality of care control, “A guideline arises from how we can control the quality of work…” * Better control and optimization of business processes, “The purpose of guidelines is to introduce order into a healthcare system. This is to be achieved indirectly by integrating CPGs into digital healthcare information systems, as such integration requires that business processes are defined.” * Better cost control, “At the end, the aim of a guideline implementation is to achieve target savings.” * Control over patients’ billing [while commenting, billing of patients who misused the ER services], “… once we implement the guidelines, we will have some control over it, and this problem should be solved.” *

A topic or a quote demonstrate: *—misconception, or **—overconfidence. For clarification please see the main text Section 3.2.3 and Section 3.4.

## Data Availability

All relevant data are within the manuscript and its Appendix A.

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
