# Peer review of "Factors Associated with the Quality and Transparency of National Guidelines: A Mixed-Methods Study"

_ijerph, 2022, doi:10.3390/ijerph19159515_

Round 1
Reviewer 1 Report
It's my pleasure to review your manuscript.
As you metioned, identifying factors that systematically affect the overall quality of national guidelines and, consequently, the quality of health care in a country is of both national and international importance. You assess the methodological quality and transparency of all CPGs published in Croatia until 2017 and to identify factors affecting a guideline development process.
There are some comments.
1) In Section Introduction, you should clearly point out the contribution to the research theme or topic.
2) I don't understand why do you only assess the methodological quality and transparency of all CPGs published in Croatia until 2017. Since 2017,has quality and transparency of all CPGs published in Croatia not changed?
Author Response
1) In Section Introduction, you should clearly point out the contribution to the research theme or topic.
We appreciate Reviewer 1's thoughtful comments on the manuscript. As a result of this suggestion, we have revised a text in the Introduction section to emphasize more the relevance of the research, the current state of the literature on factors affecting the quality of guidelines, and our contribution to the field (text in red).
2) I don't understand why do you only assess the methodological quality and transparency of all CPGs published in Croatia until 2017. Since 2017 has quality and transparency of all CPGs published in Croatia not changed?
The inclusion up to 2017 was made for purely operational reasons. Our deceased rater had health issues starting in 2018, and given that the majority of the work had already been completed, we did not want to introduce bias by inviting a new rater on the newest guidelines. We have, however, reflected on this issue in the limitation section: “We examined the guidelines up to 2017. Given that the guideline framework has not changed in the meantime, and that the Coronavirus Disease 2019 (COVID-19) pandemic temporarily impacted GD activity from 2020, we believe that the data reported accurately represent the state of Croatian guidelines in a pandemic-free setting. Considering that a 14-year time span had little effect on the quality of CPGs in Croatia, and that the guideline framework has not changed, we do not expect CPGs published in pre-pandemic 2018 (N=6) and 2019 (N=8) to have a significant impact on results.”
Reviewer 2 Report
The paper is interesting and well done. It is written in a great English and the exposition of contents is clear. The appendix explains fully all steps of used methodology. One remarks is related to the lack of reference to the comparison with other countries in the description of methodology. Maybe it could be useful a better organization of the tables and figures distribution in the text. The practical suggestions in discussion and in conclusions are really apprecciated and coherent between them.
Author Response
The paper is interesting and well done. It is written in a great English and the exposition of contents is clear. The appendix explains fully all steps of used methodology.
We thank the Reviewer 2 for these thoughtful comments on the manuscript.
One remarks is related to the lack of reference to the comparison with other countries in the description of methodology.
We thank the Reviewer for this remark. We have now added the description of methods used for comparison with other countries in the Methodology section as well as in the Section 1 of the Supplementary file (Methods - Quantitative research). Also, based on the Reviewer 3's comment, we now only compare data from countries where more recent guidelines have been studied (at least some of the guidelines analysed should have been published within the last 10 years). We also improved our search filter to explicitly include every country in the world and identified data from two new countries: China and Peru. Nevertheless, even with these adjustments (excluding Argentina but including China and Peru) our observations remain robust.
Maybe it could be useful a better organization of the tables and figures distribution in the text.
We thank the Reviewer for this comment. Tables and figures in the text have been reorganized.
The practical suggestions in discussion and in conclusions are really apprecciated and coherent between them.
We greatly appreciate this thoughtful comment by the Reviewer 2.
Reviewer 3 Report
This study employs a mixed method approach to explore the methodological quality and transparency of all national clinical practice guidelines published in Croatia until 2017. Evaluating guidelines using a validated AGREE II instrument, multiple linear regression, and two focus groups, this study has found that the majority of guidelines were developed by medical societies with poor quality. Inadequate methodology, lack of implementation systems in place, lack of awareness about editorial independence, and broader expertise/perspectives in working groups were identified as factors behind low-scores. This manuscript was generally well written and the methods were conducted adequately, I have several suggestions that could be considered to strengthen the contributions of this manuscript.
In the introduction, the authors have pointed out that the majority of guidelines are of low quality, but you do not present reasons/factors that influence the quality of these guidelines. It would also be helpful to provide a review on what factors have systematically affected the overall quality of national guidelines.
The authors have mentioned the descriptive table in the manuscript, but it seems you do not present this table?
As for the regression table, I suggest the authors including R2.
For the cross-country comparison, it seems that the authors picked up one study from each country and compared their findings with findings from current study. I am not sure of the quality of that particular study selected for each country. Moreover, the time frame is also different. The guidelines were published a long time ago in some countries (e.g. Argentina). The results of the country comparison should be interpreted with caution.
It seems to me the discussion is a summary of the findings. How do you relate what you have found to existing literature? How do your findings advance our current understanding on the development of guidelines?
I appreciate the authors’ presenting the “additional observations”. I feel this section is weakly associated with the study aim. You may consider moving this section into the appendix.
Minor
Some editorial issues should be attended to. For example, the page numbers in the manuscript are not ordered properly. The reference items are numbered twice.
Author Response
In the introduction, the authors have pointed out that the majority of guidelines are of low quality, but you do not present reasons/factors that influence the quality of these guidelines. It would also be helpful to provide a review on what factors have systematically affected the overall quality of national guidelines.
We thank the Reviewer 3 for this comment. We have now revised the text in the Introduction section to emphasize more clearly the relevance of the research, the current state of the literature on factors affecting the quality of national guidelines, and our contribution to the field (text in red). As stated in the Introduction, there is a scarcity of literature on the subject, and this study is the first to examine in depth the factors that influence the quality of national guidelines, employing both a quantitative and qualitative approach.
The authors have mentioned the descriptive table in the manuscript, but it seems you do not present this table?
We looked for the reference to a descriptive table in the main text but were not able to find it. If the Reviewer is referring to description of guideline characteristics, descriptive information is provided in the text on page 5. Given that the information isn't overwhelming and that we are commenting on it in a text, we'd prefer to present it this way.
As for the regression table, I suggest the authors including R2.
The value is now added in the Table 1 and marked in red.
For the cross-country comparison, it seems that the authors picked up one study from each country and compared their findings with findings from current study. I am not sure of the quality of that particular study selected for each country. Moreover, the time frame is also different. The guidelines were published a long time ago in some countries (e.g. Argentina). The results of the country comparison should be interpreted with caution.
We thank the reviewer for this comment. In fact, our original search for the studies was done systematically using the controlled Mesh dictionary and key words countr* and region* in the title and abstract. The description of methods used for comparison with other countries is now added in the Methodology section as well as in the Section 1 of the Supplementary file (Methods - Quantitative research). Prompted by the Reviewer’s comment, we have additionally expanded our search filter to explicitly include every country in the world and have identified studies from two new countries: China and Peru. Also, in line with the reviewer’s comment, we now only compare data from countries where more recent guidelines have been studied (at least some of the guidelines should have been published within the last 10 years). Nonetheless, even after these adjustments (excluding Argentina and including China and Peru), our findings remain robust. The difference is that, while the "Clarity-of-Presentation" and "Scope-and-Purpose" domains continue to have the highest score, these domains receive low-quality ratings in countries producing CPGs with extremely low overall AGREE II scores (<30%).
It seems to me the discussion is a summary of the findings. How do you relate what you have found to existing literature? How do your findings advance our current understanding on the development of guidelines?
We thank the Reviewer for this comment. As pointed out earlier, there is a scarcity of literature on the subject, and this study is the first to examine in depth the factors that influence the quality of national guidelines, employing both a quantitative and qualitative approach. We thus relate to the existing literature for each finding separately. In the Conclusion section, we summarized how our findings contribute to current understanding of guidelines development in a country that produces low-quality CPGs (e.g. guideline framework itself is not enough to guarantee quality; medical societies are likely not an ideal choice of organization for developing guidelines; inadequate methodology as well as the lack of methodological knowledge, misconceptions and overconfidence among guideline developers, may significantly affect the quality of CPGs, etc).
I appreciate the authors’ presenting the “additional observations”. I feel this section is weakly associated with the study aim. You may consider moving this section into the appendix.
We appreciate the Reviewer's input. While the title Additional observations may appear to be unrelated to the study's goal, we believe that these topics, which emerged during the focus group discussions and demonstrate lack of methodological knowledge, misconceptions and overconfidence among guideline developers and users; are important. As a result, we would prefer to keep the section in the main text. We have also renamed this section into: Methodological knowledge, misconceptions, and overconfidence among guideline developers and users; to emphasize its significance.
Minor
Some editorial issues should be attended to. For example, the page numbers in the manuscript are not ordered properly. The reference items are numbered twice.
This is now corrected in the text.
Round 2
Reviewer 1 Report
Thanks. The revised manuscript has made significant improvement, so I think it should be accepted in IJERPH.
This manuscript is a resubmission of an earlier submission. The following is a list of the peer review reports and author responses from that submission.